# Life-Cycle Assessment in the LEED-CI v4 Categories of Location and Transportation (LT) and Energy and Atmosphere (EA) in California: A Case Study of Two Strategies for LEED Projects

Svetlana Pushkar 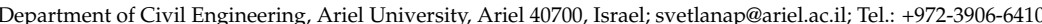

Department of Civil Engineering, Ariel University, Ariel 40700, Israel; svetlanap@ariel.ac.il; Tel.: +972-3906-6410

**Abstract:** This study aimed to identify different certification strategies for Leadership in Energy and Environmental Design Commercial Interior version 4 (LEED-CI v4) gold-certified office projects in California's cities and to explore these certification strategies using life-cycle assessments (LCAs). The LEED-CI v4 data were divided into two groups: high- and low-achievement groups in the Location and Transportation (LT) category. The author identified two strategies for achieving the same level of certification across LEED-CI v4 projects: (1) high achievements in LT ($LT_{High}$) and low achievements in the Energy and Atmosphere (EA) category ($EA_{Low}$), and (2) low achievements in the LT category ($LT_{Low}$) and high achievements in EA ($EA_{High}$). The author adopted $LT_{High}$–$EA_{Low}$ and $LT_{Low}$–$EA_{High}$ achievements as functional units for LCA. Three alternatives were $LT_{High}$: typical bus, $EA_{Low}$: gas; $LT_{Low}$: typical car, $EA_{High}$: gas; and $LT_{Low}$: eco-friendly car, $EA_{High}$: gas, where a typical bus used diesel, a typical car used natural gas, an eco-friendly car used EURO5diesel, and natural gas was used as a building's operational energy. The ReCiPe2016 results showed that the $LT_{High}$: typical bus, $EA_{Low}$: gas strategy was preferable from a short-term perspective, and the $LT_{Low}$: eco-friendly car, $EA_{High}$: gas strategy was preferable in a long-term and an infinite time perspective, while the $LT_{Low}$: typical car, $EA_{High}$: gas strategy continued to be the most environmentally damaging certification strategy for all the time horizons of the existing pollutants. Thus, it can be concluded that if there are alternative strategies for LEED certification, an analysis of their LCAs can be useful to refine the best sustainable strategy.

**Keywords:** California; LEED certification; LCA; location and transportation credits; energy and atmosphere credits; ReCiPe2016 method

## 1. Introduction

### 1.1. Problem Statement

Leadership in Energy and Environmental Design (LEED) is one of the most popular US-based building rating systems and is also known as an international sustainable tool. LEED contains the credits' requirements, organized in eight environmental categories that deal with transport, sites, water, energy, materials, indoors, innovation, and regional issues. The credits have different weightings, reflecting their environmental importance. Such a weighting set is designed by a stakeholder group of environmental specialists and building practitioners (the "stakeholder approach"). The group decides on a country-specific list of the environmental categories and the total number of points awarded. Then, the points are divided among categories according to their importance. Eventually, the total number of category points is redistributed among the credits of this category. This is carried out according to the category/credit importance decided by a stakeholder group.

Over the decades, LEED has been criticized for its subjective approach to dividing awarded points among the categories and credits [1] and delinking LEED performance from life-cycle assessment (LCA) outcomes [2]. LCA is a methodology that was created

by ISO 14040 [3] to evaluate environmental impacts and damage resulting from the whole life cycle of the project/service. Therefore, it is important to use LCA in deciding the importance of LEED credits.

In respect of linking LEED performance to LCA outcomes, LEED v4 for Building Design and Construction (BC + D) has already included building life-cycle impact reduction credits in the material and resources (MR) category [4]. This is a good starting point. However, as seen in the literature, linking the LEED system to LCA outcomes has not been completed yet. This study aimed to continue exploring this problem by linking LEED certification strategies to LCA outcomes. To find the particular gaps in this research topic, LEED certification studies and linking LEED certification to LCA outcomes studies are discussed in Sections 1.2 and 1.3, respectively, and the goals of this study are discussed in Section 1.4.

### 1.2. LEED Certification

LEED-certified projects have different certification strategies depending on the country's location, certification level, project type, and project size, and, therefore, much research has been published on this matter. For example, Wu et al. [5] collected a total of 3416 LEED for New Construction (LEED-NC) v3 2009 projects from the USA (2770 projects), China (126 projects), Turkey (53 projects), Brazil (40 projects), Chile (34 projects), and Germany (30 projects). The authors pooled all the projects together in one set and sorted them into four certification levels. As a result, at the certified level, there were 655 projects; at the silver level, there were 1310 projects; at the gold level, there were 1201 projects; and at the platinum level, there were 244 projects. Wu et al. [5] (p. 375) used the mean $\pm$ standard deviation (SD) and coefficients of variation (CV), where CV = SD/mean. For example, the mean $\pm$ SD and CV of the energy and atmosphere (EA) category at the certified level was CV = 0.53; at the silver level, CV = 0.44; at the gold level, CV = 0.36; and at the platinum level, CV = 0.17. As a result, the CV value monotonically decreased from the certified level to the platinum through the silver and gold levels. The CV values of the EA category (35 points) in LEED for newly constructed LEED-NC v3-certified, silver, gold, and platinum projects were 0.53, 0.44, 0.36, and 0.17, respectively. As a result, decreasing the degree of variation in the points achieved in the EA category can be associated with a decreasing degree of variation in the other LEED categories: sustainable sites, SS (26 points); indoor environmental quality, EQ (15 points); materials and resources, MR (14 points); and water efficiency, WE (10 points), and vice versa. The implementation of these two possible tendencies can be converted into a different LEED strategy for the achievement of LEED certification levels. According to data collected by Wu et al. [5], the revealed dependence reflects LEED projects from the US rather than other countries. However, the US LEED data are not homogeneous, as green building policies such as ASHRAE 90.1 (the Energy Standard for Buildings Except Low-Rise Residential Buildings) are determined on a state-by-state basis [6].

In the next three publications [7–9], LEED data were analyzed using the median and interquartile range (IQR, 25–75th percentiles). Therefore, instead of SD/mean, the IQR/median ratio was calculated.

Pushkar and Verbitsky [7] analyzed LEED-NC v3 2009 gold projects certified in 2016 in several US states such as California (CA = 58 projects), Illinois (IL = 19 projects), Florida (FL = 11 projects), Washington (WA = 11 projects), Ohio (OH = 8 projects), and Massachusetts (MA = 14 projects). As a result, in the EA category, the following two sub-groups were revealed: (1) high values of the IQR/median ratio and (2) low values of the IQR/median ratio. The first group included CA, 22.0 $\pm$ 13.0 (0.59); IL, 17.0 $\pm$ 14.5 (0.85); and FL, 15.0 $\pm$ 11.0 (0.73), while the second group included WA, 16.0 $\pm$ 4.3 (0.27); OH, 13.5 $\pm$ 4.0 (0.30); and MA, 14.0 $\pm$ 5.0 (0.36). In this context, under the same gold certification, at least two facts are notable: (1) there were states with different LEED strategies (high values of the IQR/median ratio), and (2) there were states with the same LEED strategy (low values of the IQR/median).

Pushkar and Verbitsky [8] (p. 98) evaluated the IQR/median ratio for LEED-NC v3 2009 gold projects certified in California in 2012–2017. These authors showed that, in the EA category, the minimum IQR/median ratio was 0.31 in 2012, and the maximum IQR/median ratio was 0.81 in 2017. In this context, in 2012, LEED strategies had low variance, while in 2017, LEED strategies had high variance when the LEED-NC v3 2009 projects had the same gold certification.

Pushkar [9] studied the difference between Shanghai and California in terms of LEED for commercial interiors (LEED-CI v4) of gold-certified office space projects. The author [9] (p. 34) showed that there was a significant difference in the IQR/median ratios in the two highest-scoring categories (location and transportation (LT), 18 points and EA, 35 points). For Shanghai, the median and (the IQR/median ratio) for LT and EA were 17.0 and (0.06) and 15.0 (0.27), respectively, while for California, (the median and the IQR/median ratio) for LT and EA were 15 (0.88) and 24 (0.55), respectively. In this context, it can be assumed that Shanghai's projects used the LT–EA certification strategies with low variation, while California's projects used the LT–EA certification strategies with high variation. It should be noted that, in both Shanghai and California, the same gold certification was analyzed.

Thus, the three articles listed above show that, for one level of certification, there can be different strategies used to achieve this.

### 1.3. Linking LEED Certification to LCA Outcomes

There have been some attempts to integrate LCA into LEED in the literature. Scheuer and Keoleian [10] evaluated solid waste generation and life-cycle energy consumption in a six-story University of Michigan campus building that resulted from a simulated application of LCA to LEED-NC v2 credits. Three MR and three EA credits out of a total of sixty-four credits were analyzed. The studied MR credits were MRc2, construction waste management; MRc4, recycled materials; and MRc5, local/regional materials, and the evaluated EA credits were EAc1, optimizing energy performance; EAc2, renewable energy; and EAc6, green power. The authors reported on a variety of discrepancies between the LEED-NC v2 rating system, in which all the credits were awarded one point, and there were completely different LCA results for these credits.

Humbert et al. [11] directly evaluated LCA outcomes from a simulated application of LEED-NC v2.2, with 45 quantifiable credits out of a total of 69 credits awarded to an actual Californian office building. The evaluated credits belonged to the SS, WE, EA, and MR categories. The authors concluded that most LEED-NCv2.2 credits brought about environmental benefits. However, several credits, such as SSc4.3, alternative transportation, low-emission and fuel-efficient vehicles and SSc7.1, heat island effect, non-roof, caused environmental damage. Moreover, Humbert et al. [11] pointed out significant discrepancies between (i) the low number of points awarded in the rating system and the high benefit of the LCA results of certain credits such as EAc6, green power, and (ii) the high number of points awarded in the rating system and the low benefit of LCA results of certain credits such as WEc1.1, water-efficient landscaping, which was reduced by 50%, for example.

Other studies also criticized the delinking of LCA from LEED certification. For example, Suh et al. [12] studied the application of 38 quantifiable LEED-NC v3 credits belonging to the SS, WE, EA, and MR categories for a prototypical, small office building consuming 6 terajoules (TJs) of primary energy and releasing about 18,000 tons $CO_2$-eq. of greenhouse gases (GHGs) over its life cycle based on national average values. It was concluded that the environmental impact reduction potentials of the LEED building simulation were unevenly distributed across the measured impacts. The largest reductions were noted for acidification (25%), human respiratory health (24%), and global warming (22%), while no reductions were observed for ozone layer depletion and land use.

Al-Ghamdi and Bilec [13] studied the building energy use and associated life-cycle impacts of typical office buildings located in 400 cities worldwide regarding their satisfaction with the LEED-NC v3 operational energy criteria based on the American Society of Heating, Refrigerating, and Air-Conditioning Engineers (ASHRAE) 90.1 energy code. The authors

reported wide variations in $CO_2$ emissions, from 394 tons $CO_2$-eq. to 911 tons $CO_2$-eq. They concluded that there is a need to consider the LCA of local-based operational energy results in order to gain a better understanding of the possible environmental impacts in the context of the energy requirements of green building rating systems.

Thus, LEED BC + D v4 included the building life-cycle impact reduction credit in the MR category [4]. This credit presents the option of receiving three points for decreasing three of six environmental impacts (global warming potential, depletion of the stratospheric ozone layer, acidification of land and water sources, eutrophication, formation of tropospheric ozone, and depletion of nonrenewable energy resources). The proposed design should reduce the impact by 10 percent compared to a baseline design built according to ASHRAE 90.1-2010. In this way, LEED v4 BC + D linked LCA to LEED certification. However, the credit's intent is to promote the reuse and optimization of building construction materials. This is a good starting point for LCA penetration into LEED systems. However, LCA is still not considered across other quantifiable LEED categories, taking into account that some EQ credits (daylighting, thermal comfort, or quality views) cannot be analyzed with the current LCA methodology.

In this respect, Greer et al. [2] evaluated LCAs of the application of three LEED BC + D v4 credits: optimizing energy performance (EA category), indoor water use reduction, and outdoor water use reduction (the WE category) in different cities of California. The authors revealed great variability in the avoided carbon dioxide, with 0.1–0.9 and 0.1–0.2 kg $CO_2/m^2/y$ per each EA and WE point awarded, respectively, in different cities of California. The $CO_2$ variability was due to different building types, electricity fuel sources for buildings' operational heating and cooling energy, and water infrastructure in this state.

### 1.4. Goals of the Study

The state of the art shows that there is still a lack of studies that link LEED certification strategies to LCA outcomes. In particular, LCAs of different LEED certification strategies have not yet been performed. The author of the present study decided to focus on LEED-CI v4 gold-certified office projects in cities in California, for which two hypotheses were elaborated: (1) based on Pushkar's study [9], it was supposed that there are different LT–EA certification strategies in California, and (2) based on Greer et al.'s study [2], it was supposed that the different LT–EA certification strategies can lead to different LCA outcomes. Thus, the first aim of this study was to reveal existing certification strategies of LEED-CI v4 gold-certified office projects in cities in California, and the second aim was to evaluate the different certification strategies via LCA outcomes.

The results of this study provide the first LCA evidence from different certification strategies that were applied by LEED building practitioners toward achieving the same certification level. In this way, the outcome of the study may help LEED experts to make further improvements to the LEED system for the additional mitigation of the environmental impacts caused by the construction sector.

## 2. Materials and Methods

### 2.1. Design of the Study

To reduce the impact of unknown factors, the author collected LEED-CI v4 office projects from California only because, in the USA, green building policies are regulated differently in each state [6]. The author selected California as a case study due to the following reasons. First, California has the largest number of LEED-CI v4-certified office projects as compared to the other US states and so is acceptable for statistical analysis [6]. Second, California's cities have completely different percentages of people using public transportation, which allowed us to assess the impact of transportation on the LEED strategy; for example, the percent of people using public transportation in San Francisco is 34.7, but in Sunnyvale, it is 7.6 [14].

Figure 1 shows a flowchart of the methodology used in the present work. The following steps were performed:

(1)  Filtering LEED-CI v4-certified, silver, gold, and platinum projects by sample size and sorting these by LT points resulted in the selection of the most appropriate gold project groups with high and low achievements in the LT category (i.e., $LT_{High}$ and $LT_{Low}$) (Section 2.2.1);

(2)  Distribution of the $LT_{High}$ and $LT_{Low}$ gold projects by cities in California and comparing them to the percentage of people using public transportation in these cities (Section 2.2.2);

(3)  Comparing the LEED certification achievements of the $LT_{High}$ projects and the $LT_{Low}$ projects by category (IP, LT, WE, EA, MR, EQ, IN, and RP) and credit levels resulted in two different LT–EA certification strategies: $LT_{High}$–$EA_{Low}$ and $LT_{Low}$–$EA_{High}$ (Section 3.1.1);

(4)  Adopting $LT_{High}$–$EA_{Low}$ and $LT_{Low}$–$EA_{High}$ achievements as a functional unit (FU) for LCA evaluations by converting the $LT_{High}$–$EA_{Low}$ LEED points into bus (typical bus) transportation distance (km) and building operational energy (OE) for heating and cooling (kWh) and converting $LT_{Low}$–$EA_{High}$ LEED points into car (typical car or eco-friendly car) transportation distance (km) and building OE for heating and cooling (kWh) (Section 2.3.1);

(5)  Evaluating the midpoint impact and endpoint single-score damage results of the $LT_{High}$–$EA_{Low}$ ($LT_{High}$: typical bus, $EA_{Low}$: gas) and $LT_{Low}$–$EA_{High}$ ($LT_{Low}$: typical car, $EA_{High}$: gas and $LT_{Low}$: eco-friendly car, $EA_{High}$: gas) certification strategies using ReCiPe2016 life-cycle impact assessment methodology (Section 3.2).

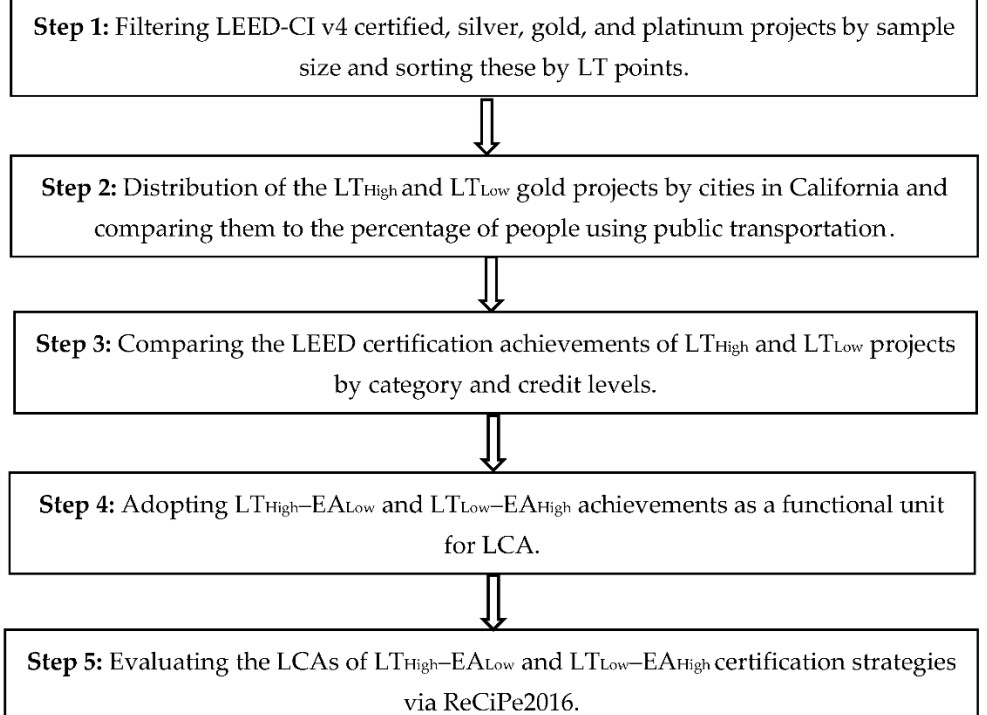

**Figure 1.** A flowchart of the methodology for five-step LCA evaluation procedure of $LT_{High}$–$EA_{Low}$ and $LT_{Low}$–$EA_{High}$ certification strategies of LEED-CI v4 gold office projects.

## 2.2. Data Collection

2.2.1. Filtering by Sample Size and Sorting by Location and Transportation (LT) Points

A total of 101 LEED-CI v4-certified, silver, gold, and platinum office projects in California (20 certified, 36 silver, 40 gold, and 5 platinum) were discovered between March 2015 and February 2022 from the following two databases: the USGBC [15] and the Green Building Information Gateway (GBIG) [16]. The USGBC database was used to collect the credit achievements in LEED-CI v4 projects, and the GBIG database was used to collect LEED-CI office projects only [16]. As can be seen from Table 1, for the certified, silver, and platinum levels of certification, the author selected three groups: high performance in the LT category, low performance in the LT category, and intermediate performance in LT category. For the gold level of certification, the author selected four groups: high performance in the LT category, low performance in the LT category, and two groups with intermediate performance in the LT category.

**Table 1.** Distribution of the LEED-CI v4 office projects according to the location and transportation (LT) category performance and levels of certification in California. Table 1 lists the four groups (certified, silver, gold, and platinum), the three LT performance levels (low, medium, and high), and the number of LEED projects for each combination of group certification and LT performance level. The gold group includes two levels of intermediate indicators: intermediate-low and intermediate-high. The LT achievement scores and the number of LEED projects were found in the USGBC and GBIG databases [15,16].

| Certified group | Low performance in $LT_{Low}$ (2–3 points) | Intermediate performance in LT (4–12 points) | | High performance in $LT_{High}$ (15–17 points) |
|---|---|---|---|---|
| Number of projects | 5 | 6 | | 9 |
| Silver group | Low performance in $LT_{Low}$ (0–5 points) | Intermediate performance in LT (8–14 points) | | High performance in $LT_{High}$ (15–18 points) |
| Number of projects | 12 | 13 | | 11 |
| **Gold group** | **Low performance in $LT_{Low}$ (1–3 points)** | **Intermediate-low performance in LT (4–8 points)** | **Intermediate-high performance in LT (14–15 points)** | **High performance in $LT_{High}$ (17–18 points)** |
| **Number of projects** | **12** | **5** | **9** | **14** |
| Platinum group | Low performance in $LT_{Low}$ (9 points) | Intermediate performance in LT (15 points) | | High performance in $LT_{High}$ (17–18 points) |
| Number of projects | 1 | 1 | | 3 |

Notes: Bold font = data evaluated in this study.

It should be noted that the LEED data contain three types of data: binary, ordinal, and discrete interval variables with relatively few values. In this context, to compare the differences between the two groups, the minimum number of LEED projects or sample size (*n*) in each group must be $n \geq 12$ [17]. According to the LEED-CI v4 office project numbers in each group, this study focused on comparing two strategies used to achieve the gold level of certification in LEED-CI v4 office projects, namely, high and low performance in the LT category (i.e., $LT_{High}$ and $LT_{Low}$).

2.2.2. Distribution of LEED-CI v4 Gold-Certified Office Projects in Cities of California According to $LT_{High}$ and $LT_{Low}$ Achievement

Table 2 shows that the $LT_{High}$ group included 13 projects from San Francisco and 1 project from Los Angeles; the $LT_{High}$ group occurred in cities with relatively high usage of public transportation, while the $LT_{Low}$ group occurred in cities with relatively low usage of public transportation.

**Table 2.** Distribution of LEED-CI v4 gold-certified office projects in cities of California according to $LT_{High}$ and $LT_{Low}$ via the percentage of people using public transportation. Table 2 provides more details on the $LT_{Low}$ and $LT_{High}$ groups by California cities for LEED gold-certified projects from Table 1. The second column represents the percentage of residents that use public transport. The last row of Table 2 contains the total number of LEED projects for each LT performance group.

| City | Percent Using Public Transportation [a] | Number of LEED Projects | |
| --- | --- | --- | --- |
| | | $LT_{Low}$ (1–3 Points Achieved) [b] | $LT_{High}$ (17–18 Points Achieved) [b] |
| San Francisco | 34.7 | – | 13 |
| Los Angeles | 8.9 | – | 1 |
| Mountain View | 8.8 | 1 | – |
| Fremont | 8.1 | 1 | – |
| Sunnyvale | 7.6 | 5 | – |
| San Diego | 4.0 | 1 | – |
| Roseville | No data | 1 | – |
| Rancho Cordova | No data | 1 | – |
| Menlo Park | No data | 1 | – |
| Brisbane | No data | 1 | – |
| Total Number of LEED Projects | | 12 | 14 |

Notes: [a] [14] and [b] [15].

### 2.3. Life-Cycle Assessment

#### 2.3.1. Functional Unit

Following the statistical evaluation of the LEED achievements of the $LT_{High}$ and the $LT_{Low}$ projects, two different certification strategies, $LT_{High}$–$EA_{Low}$ and $LT_{Low}$–$EA_{High}$, were revealed. LTc3 (access to quality transit) and EAc6 (optimize energy performance) were determined to be representative credits of $LT_{Hight}$ or $LT_{Low}$ and $EA_{High}$ or $EA_{Low}$ for LCA evaluation (see the detailed explanation in Section 3.1.1).

For the comparison of the LCAs of the $LT_{High}$–$EA_{Low}$ and $LT_{Low}$–$EA_{High}$ certification strategies, both transportation (LTc3) and operational energy (EAc6) need to be considered as a single FU. Therefore, the FU was designated as follows: one passenger-related transportation from home to office and back + 8 h of OE service for one employee. In particular, this FU included the transport of one employee per 30 km of distance (LTc3) + OE per one employee per 20 $m^2$ office space (EAc6) per one day of office work.

The FU uses 30 km of distance per one employee due to the reported average traveling distance from home to work for California [18] and 20 $m^2$ of office space per one employee as a common maximum operational energy design criteria for an office-type building [19]. An office employee traveling from home to work was assumed to travel by bus for $LT_{High}$ and by car for $LT_{Low}$. In the case of traveling by car, the author of this study considered two options: the current situation, in which California's cars are fueled by natural gas, and a hypothetical future situation, in which California could become a leader in the development of new, more environmentally friendly transport [20].

$EA_{Low}$ and $EA_{High}$ refer to the quantity of kWh for the OE used by each of the LEED-CI v4 gold-certified office projects for their heating and cooling needs. As a starting point, this study adopted 80 kWh/$m^2$ of OE as a base case for California's office buildings [21]. It was assumed that the OE was produced by natural gas, which is the most common electricity generation fuel in California [22]. Then, the author performed a four-step evaluation procedure in which, for each of the analyzed projects, the LEED points were converted into kWh that were used as input data for the LCA of $EA_{High}$ and $EA_{Low}$. A description of this procedure is presented in Appendix A.

#### 2.3.2. Life-Cycle Inventory

The LCIs of the $LT_{High}$–$EA_{Low}$ and $LT_{Low}$–$EA_{High}$ certification strategies were modeled on the SimaPro platform [23]. The Ecoinvent database has a comprehensive transportation

energy-related LCI database [23]. Table 3 shows the Ecoinvent v3.2 database sources adopted for transportation (used in $LT_{High}$ and $LT_{Low}$) and OE (used in $EA_{High}$ and $EA_{Low}$).

**Table 3.** Transportation and operational energy processes: data input from the Ecoinvent v3.2 database (SimaPro v9.1 [23]).

| Process | Data Source [23] |
|---|---|
| Transportation by a typical bus | Transport, regular bus/CH S |
| Transportation by a typical car | Transport, passenger car, natural gas/CH S |
| Transportation by an eco-friendly car | Transport, passenger car, diesel, EURO5, city car/CH S |
| Operational energy (OE) | Electricity by fuel, gas, electricity, natural gas, at power plant/US S |

Notes: CH: Switzerland and US: the United States.

For OE, the author used the original US database. However, due to the absence of an original US database for transportation, it was necessary to adopt the Switzerland database, which was considered appropriate due to the comparative nature of the evaluation in the present study.

According to the Ecoinvent v3.2 database [23], transportation by a typical bus refers to the entire transport life cycle and includes bus production, operation, maintenance, and disposal, as well as the construction, renewal, and disposal of roads. A vehicle lifetime performance of 23,900 personkm/vehicle was assumed. The data for vehicle operation and road infrastructure reflect Switzerland's conditions. The data for vehicle manufacturing and maintenance represent generic European data.

The Ecoinvent v3.2 database [23] states that transportation by a typical car includes data on Euro3 vehicle operation, and bitumen and concrete comprise roads. Inventory refers to the entire transport life cycle. The vehicle manufacturing data reflect current modern technologies. It also includes the construction, renewal, and disposal of roads, as well as the operation of the road infrastructure. A vehicle's lifetime performance was 23,900 personkm/vehicle, with an average utilization of 1.59 passengers/car. The data for vehicle manufacturing and maintenance represent generic European data, whereas the data for vehicle operation and road infrastructure reflect Switzerland's conditions.

According to the Ecoinvent v3.2 database [23], transportation by eco-friendly cars (diesel cars, lightweight concept, 2l t/100 km, EURO5) takes into account an average load factor of 1.6 persons. Inventory refers to the entire transport life cycle, car production, operation, and maintenance, as well as the operation and disposal of road infrastructure. A vehicle's lifetime performance was 150,000 km/vehicle. The data for vehicle life cycle and road infrastructure reflect Switzerland's conditions.

The Ecoinvent v3.2 database [23] states that the production of electricity has an average net efficiency of 100% in the natural gas power plants in the USA. Technology reflects electricity production by natural gas steam generation.

Thus, due to Switzerland's inventory data for a typical bus, a typical car, and an eco-friendly car, the LCAs of the $LT_{High}$–$EA_{Low}$ ($LT_{High}$: typical bus, $EA_{Low}$: gas), $LT_{Low}$–$EA_{High}$ ($LT_{Low}$: typical car, $EA_{High}$: gas), and $LT_{Low}$–$EA_{High}$ ($LT_{Low}$: eco-friendly car, $EA_{High}$: gas) certification strategies are fully comparable with each other.

### 2.3.3. Life-Cycle Impact Assessment

The author of the present study used the ReCiPe2016 life-cycle impact assessment (LCIA) method. This method is based on individualist (I), hierarchical (H), and egalitarian (E) views regarding environmental problems. The individualist view accounts for a short lifetime (20 years), the hierarchical view accounts for a long lifetime (100 years), and the egalitarian view accounts for an infinite lifetime (1000 years) of pollutants [24,25].

To analyze the $LT_{High}$–$EA_{Low}$ and $LT_{Low}$–$EA_{High}$ certification strategies, this study used both midpoint (H) and endpoint single-score (individualist/average, I/A; hierarchi-

cal/average, H/A; and egalitarian/average, E/A) evaluations. On the midpoint scale, the author evaluated global warming, human carcinogenic toxicity, human noncarcinogenic toxicity, and terrestrial ecotoxicity impacts. These impacts were selected as they were the most influenced by transportation and operational energy processes [23]. Table 4 shows these impacts for 1 km (transportation) and 1 kWh (OE).

**Table 4.** Transportation and operational energy (OE) processes: environmental impacts (ReCiPe2016, hierarchical perspective) [23].

| Process | GW (kg $CO_2$ eq) | HCT (kg 1,4-DCB) | Hn-CT (kg 1,4-DCB) | TE (kg 1,4-DCB) |
|---|---|---|---|---|
| Typical bus (1 personkm) | 0.10500 | 0.00245 | 0.00232 | 0.14800 |
| Typical car (1 personkm) | 0.17000 | 0.00794 | 0.00895 | 0.20900 |
| Eco-friendly car (1 personkm) | 0.05390 | 0.00323 | 0.00471 | 0.10900 |
| OE: electricity (1 kWh) | 0.75100 | 0.00004 | 0.06210 | 0.00645 |

Notes: GW, global warming; HCT, human carcinogenic toxicity; Hn-CT, human noncarcinogenic toxicity; TE, terrestrial ecotoxicity. $CO_2$ eq is the contribution of methane ($CH_4$), nitrous oxide ($N_2O$), and carbon dioxide ($CO_2$) to GW expressed in $CO_2$ equivalent quantity.

### 2.4. Statistical Analysis

#### 2.4.1. Choice of Statistical Procedures

LEED data are expressed on ordinal or discrete interval variables with relatively few values or binary data. For descriptive statistics, this paper used the median and 25th and 75th percentiles, and for inferential statistics, nonparametric tests were used because the normality assumption may not hold [26].

For ordinal or discrete data, to estimate the *p*-value, this paper used the exact Wilcoxon–Mann–Whitney (WMW) nonparametric test [17], and to estimate the effect size, a nonparametric Cliff's $\delta$ test was used [27].

For LEED binary data, to estimate the *p*-value, this paper used Fisher's exact $2 \times 2$ test with Lancaster's mid-*p*-value [28]. To estimate the effect size, (1) the author computed odds ratios using a two-by-two frequency table, but added 0.5 to each frequency observed if any of them were 0 [29], and (2) the author used the natural logarithm of the odds ratio ($\ln \theta$) [30].

LCA–LEED data are expressed as discrete data. However, as these data were being analyzed for the first time, the author performed a Shapiro–Wilk test to estimate the assumption of normality. For $LT_{High}$: typical bus data, in each perspective (i.e., I/A, H/A, and E/A), the Shapiro–Wilk test results showed that the assumption of normality was not met ($p = 0.0008$), while for $LT_{Low}$: typical car and $LT_{Low}$: eco-friendly car data in each perspective (i.e., I/A, H/A, and E/A), the Shapiro–Wilk test results showed that the assumption of normality was met ($p = 0.0598$, $p = 0.1264$, and $p = 0.2222$), respectively. In this context, if one of the two groups does not have a normality assumption, the nonparametric exact WMW test and Cliff's $\delta$ effect size are used to estimate the statistical difference between the two groups.

#### 2.4.2. Effect Size Interpretation

Nonparametric Cliff's $\delta$ was applied to measure the effect size of the difference between the two distributions [27]. Cliff's $\delta$ ranges between $-1$ and $+1$. A positive value (+) indicates that Group 1 (i.e., $LT_{High}$) was larger than Group 2 (i.e., $LT_{Low}$); a value of 0 indicates equality or overlap (i.e., equality between groups $LT_{High}$ and $LT_{Low}$); and a negative value (−) indicates that Group 2 (i.e., $LT_{Low}$) was larger than Group 1 (i.e., $LT_{High}$). The Cliff's $\delta$ effect size is negligible if $|\delta| < 0.147$, small if $0.147 \leq |\delta| < 0.33$, medium if $0.33 \leq |\delta| < 0.474$, and large if $|\delta| \geq 0.474$ [31].

The value of $\ln \theta$ ranges between (–) infinity and (+) infinity [29]. A positive value indicates that Group 1 (i.e., $LT_{High}$) was larger than Group 2 (i.e., $LT_{Low}$); a value of 0 indicates no difference between Groups 1 and 2 (i.e., no difference between groups $LT_{High}$ and $LT_{Low}$); and a negative value indicates that Group 2 (i.e., $LT_{Low}$) was larger than

Group 1 (i.e., $LT_{High}$). The effect size thresholds of the absolute $\ln\theta$ ($|\ln\theta|$) were 0.51 (small), 1.24 (medium), and 1.90 (large) and were adapted from the study by Chen et al. [32].

According to Altomonte [33], the Cliff's $\delta$ coefficient is an intuitive interpretation of the practical significance (i.e., effect size) in green building studies. This is likely due to the small number of studies in this area that have used effect size coefficients. Vargha and Delaney [34] noted that more empirical evidence is needed to evaluate the real effect size for nonparametric group comparisons.

### 2.4.2.1. *p*-Value Interpretation

According to Hurlbert and Lombardi [35], exact *p*-values are evaluated according to a three-valued logic: seems to be positive (i.e., there seems to be a difference between Group 1 and Group 2), seems to be negative (i.e., there does not seem to be a difference between the groups), or judgment is suspended regarding the difference between Groups 1 and 2. Recently, the author of [26] described the interpretation of the *p*-value in more detail.

### 3. Results

### *3.1. Preliminary Results*

### 3.1.1. LEED Certification Achievements of the $LT_{High}$ and the $LT_{Low}$ Projects

Table 5 gives descriptive and inferential statistics for the categories of the LEED-CI v4 gold certification. According to the LEED total, both the $LT_{Low}$ and $LT_{High}$ certification strategies led to similar achievements of total median points, 62.5 and 63.0, respectively. However, similar achievements were recorded for both similarly achieved categories, such as IP, WE, MR, and RP, and differently achieved categories, such as LT, EA, EQ, and IN.

**Table 5.** LEED-CI v4 gold-certified projects in California: $LT_{Low}$ versus $LT_{High}$ achievements.

| Category | Possible Points | Median, 25–75th Percentiles | | $\delta$ | *p*-Value |
| --- | --- | --- | --- | --- | --- |
| | | $LT_{Low}$ | $LT_{High}$ | | |
| Integrative process (IP) | 2 | 1.0 0.5–2.0 | 2.0 0.0–2.0 | −0.15 | 0.4797 |
| Location and transportation (LT) | 18 | 3.0 2.0–3.0 | 17.0 17.0–18.0 | −1.00 | **0.00001** |
| Water efficiency (WE) | 12 | 8.0 7.0–10.0 | 6.0 6.0–8.0 | 0.39 | 0.0991 |
| Energy and atmosphere (EA) | 38 | 27.5 24.5–30.5 | 16.0 14.0–22.0 | 0.82 | **0.0001** |
| Materials and resources (MR) | 13 | 5.5 5.0–6.5 | 5.0 5.0–6.0 | 0.26 | 0.2720 |
| Indoor environmental quality (EQ) | 17 | 9.0 8.0–9.5 | 7.0 6.0–8.0 | 0.49 | **0.0282** |
| Innovation (IN) | 6 | 6.0 5.5–6.0 | 5.0 4.0–6.0 | 0.54 | **0.0169** |
| Regional priority (RP) | 4 | 3.0 3.0–4.0 | 3.0 3.0–4.0 | −0.02 | 1.0000 |
| LEED total | 79 | 62.5 61.0–64.5 | 63.0 60.0–64.0 | 0.01 | 0.9687 |

Notes: *p*-values were evaluated according to three-valued logic; bold font indicates that the value seems to be positive; Roman font indicates that the value seems to be negative.

Among the differently achieved categories, LT (which emphasized the preferability of public transportation) performed better in the $LT_{High}$ group of the projects than in the $LT_{Low}$ group of the projects. Such results were expected. This is because, in the $LT_{High}$ group, 13 of the 14 LEED-CI v4 gold projects were certified in San Francisco (Table 2), the densest city with a highly developed public transportation system [20]. With regard to the $LT_{Low}$ group of projects, they were certified in other Californian cities such as Brisbane, Fremont, Menlo Park, Mountain View, Rancho Cordova, Roseville, San Diego, and Sunnyvale (Table 2). In this respect, Turrentine [20] noted that, unless Californians live in San Francisco, "they also are likely to have never carpooled with their neighbors, despite the presence of High Occupancy Vehicle lanes on many California freeways, and they have probably never used mass transit".

Thus, to compensate for low LT achievement, the $LT_{Low}$ group of projects was forced to receive more points under other categories. This phenomenon of the interdependence between LEED categories' achievements was described early on by Ismaeel [36], who developed a dynamic model for sustainable site selection according to LEED-NC v4. The author revealed that as the achievements of the site selection categories (LT and SS) decreased, the achievements of the WE, EA, MR, and EQ categories increased. Ismaeel [36] concluded that the highest influence of the site selection categories was for EA credits, followed by EQ, MR, and WE. This was explained by the fact that when site selection categories have local constraints (e.g., lack of public transportation), LEED practitioners are forced to aim for higher performance in other categories.

In the present study, to compensate for low achievement, the $LT_{Low}$ group of projects invested in improving the EA, EQ, and IN categories. As a result, in these three categories, the $LT_{Low}$ group of projects had better achievements than the $LT_{High}$ group (Table 5). However, it is not clear how LT is related to some of the EQ credits (e.g., daylighting thermal comfort or quality views) and IN credits (they can be completely different in different projects). Thus, the EQ and IN categories were outside the scope of this study.

In this respect, Table 6 gives descriptive and inferential statistics only for the LT and EA credits of the LEED-CI v4 gold certification. For the three LT credits, LTc2 (surrounding density and diverse uses), LTc3 (access to quality transit), and LTc5 (reduced parking footprint), the $LT_{High}$ group of projects received the maximum possible points, significantly outperforming the $LT_{Low}$ group. However, for three EA credits, EAc2 (advanced energy metering), EAc4 (enhanced refrigerant management), and EAc6 (optimize energy performance), the $LT_{Low}$ group of projects outperformed the $LT_{High}$ group.

**Table 6.** LEED-CI v4 gold-certified projects in California: location and transportation (LT) and energy and atmosphere (EA) credits.

| Credit | Possible Points | Median, 25–75th Percentiles | | $\delta/\ln\theta$ | *p*-Value |
| --- | --- | --- | --- | --- | --- |
| | | $LT_{Low}$ | $LT_{High}$ | | |
| **Location and Transportation** | | | | | |
| LTc2, surrounding density and diverse uses [a] | 8 | 2.0 1.0–2.0 | 8.0 8.0–8.0 | −1.00 | **0.00001** |
| LTc3, access to quality transit [a] | 7 | 0.0 0.0–0.0 | 7.0 7.0–7.0 | −1.00 | **0.00001** |
| LTc4, bicycle facilities [b] | 1 | 1.0 1.0–1.0 | 0.5 0.0–1.0 | 1.61 | *0.0738* |
| LTc5, reduced parking footprint [a] | 2 | 0.0 0.0–0.0 | 2.0 2.0–2.0 | −1.00 | **0.00001** |
| **Energy and Atmosphere** | | | | | |
| EAc1, enhanced commissioning [a] | 5 | 4.0 4.0–5.0 | 4.0 4.0–5.0 | 0.01 | 0.9414 |
| EAc2, advanced energy metering [a] | 2 | 1.0 1.0–2.0 | 0.0 0.0–1.0 | 0.47 | **0.0241** |
| EAc3, renewable energy production [a] | 3 | 0.0 0.0–0.5 | 0.0 0.0–0.0 | 0.25 | 0.1692 |
| EAc4 [b], enhanced refrigerant management [b] | 1 | 1.0 0.5–1.0 | 0.0 0.0–0.0 | 2.40 | **0.0121** |
| EAc5, green power and carbon offsets [a] | 2 | 0.5 0.0–2.0 | 2.0 0.0–2.0 | −0.12 | 0.6951 |
| EAc6, optimize energy performance [a] | 25 | 20.0 19.5–21.5 | 9.0 7.0–17.0 | 0.80 | **0.0002** |

Notes: *p*-values were evaluated according to three-valued logic; bold font indicates that the value seems to be positive; Roman font indicates that the value seems to be negative; italic font indicates that judgment is suspended. [a] Exact WMW test and Cliff's $\delta$ were used. [b] Fisher's exact $2 \times 2$ test and $\ln\theta$ were used.

LTc2 (surrounding density and diverse uses) deals with the presence of city infrastructure near a building site; LTc3 (access to quality transit) considers the presence of public transportation in the vicinity of a building site; LTc5 (reduced parking footprint) recommends reducing private car parking places [37]. All of these LT credits can help decrease the main fuel combustion emissions, such as nitrogen dioxide ($NO_2$) and carbon monoxide (CO), which are known to be human carcinogens with terrestrial ecotoxicity impacts [23].

EAc2 (advanced energy metering) aims to control energy savings; EAc6 (optimize energy performance) encourages insulating building envelopes and installing energy-efficient systems [37]. Using fossil fuels releases emissions such as sulfur dioxide ($SO_2$) and nitrogen

oxides ($NO_x$), thereby increasing acidification and human toxicity impacts, respectively [23]. EAc4 (enhanced refrigerant management) requires a decrease in chlorofluorocarbons (CFCs) and hydrochlorofluorocarbons (HCFCs), which can contribute to ozone depletion [37].

As can be observed, the LT and EA credits involve different areas of human health and environmental protection and, as a consequence, can decrease the different impacts associated with each of them. Thus, to obtain LEED-CI v4 gold certification, the $LT_{High}$ group of projects preferred to decrease the LT-related impacts and increase the EA-related impacts, whereas the $LT_{Low}$ group preferred to decrease the EA-related impacts and increase the LT-related impacts. Thus, the LCA of these two certification strategies was further developed, exploring them in terms of the environmental impact and damage level.

Two LT credits, LTc2 (surrounding density and diverse uses) and LTc3 (access to quality transit), are the most important due to receiving 15 out of 18 points (Table 6). However, LTc2 concerns walking, whereas LTc3 concerns driving. Thus, LTc3 will influence the environment in a much more straightforward manner than LTc2. EAc6 (optimize energy performance) is the most influential credit in the EA category due to it accounting for the greatest number of points: 25 out of 32 (Table 6). Moreover, EAc6 can be easily accounted for in the LCA framework. Eventually, the author decided to perform LCAs for LTc3 (access to quality transit) and EAc6 (optimize energy performance), as these are the most representative credits of the LT and EA categories, respectively. These credits were selected for their large influence on the LT and EA categories and the possibility of translating the credits' requirements into quantitative LCA inputs. Thus, the LCAs of two different strategies, $LT_{High}$–$EA_{Low}$ and $LT_{Low}$–$EA_{High}$, were evaluated, and the results are presented below.

### 3.1.2. LCAs of $LT_{High}$–$EA_{Low}$ and $LT_{Low}$–$EA_{High}$

Following the evaluation procedure described in Appendix A, LEED points were converted into kWh for all the projects analyzed. Tables 7 and 8 give LTc3 and EAc6 information for the LCAs of the $LT_{High}$–$EA_{Low}$ and $LT_{Low}$–$EA_{High}$ certification strategies, respectively.

**Table 7.** LEED-CI v4 gold-certified office-type projects: LTc3 and EAc6 information for LCAs of the $LT_{High}$–$EA_{Low}$ project group.

| Project Address | LTc3 [a] | EAc6 [a] | EAc6 [b] | EAc6 [b] |
|---|---|---|---|---|
| | (7 Possible Points) | (25 Possible Points) | (%) | (kWh·Day·20 m$^2$) |
| 523 W 6th St., Los Angeles | 7 | 17 | 14 | 5.5 |
| 235 Pine St., San Francisco | 7 | 8 | 6 | 6.0 |
| 350 California St., San Francisco | 7 | 10 | 7 | 6.0 |
| 111 Sutter St., San Francisco | 7 | 6 | 5 | 6.1 |
| 4 Embarcadero Center, San Francisco | 7 | 8 | 6 | 6.0 |
| 350 Rhode Island St., San Francisco | 7 | 7 | 6 | 6.0 |
| 4 Embarcadero St., San Francisco | 7 | 4 | 4 | 6.1 |
| 440 Turk St., San Francisco | 7 | 18 | 15 | 5.4 |
| 333 Valencia St., San Francisco | 7 | 17 | 14 | 5.5 |
| 1 Front St., San Francisco | 7 | 25 | 28 | 4.6 |
| 945 Bryant St., San Francisco | 7 | 15 | 12 | 5.6 |
| 1088 Sansome St., San Francisco | 7 | 10 | 7 | 6.0 |
| 1725 Third St., San Francisco | 6 | 7 | 6 | 6.0 |
| 1655 Third St., San Franco | 6 | 7 | 6 | 6.0 |

Notes: [a] [15]; [b] evaluations performed in the present study.

**Table 8.** LEED-CI v4 gold-certified office-type projects: LTc3 and EAc6 information for LCAs of the $LT_{Low}$–$EA_{High}$ project group.

| Project Address | LTc3 [a] | EAc6 [a] | EAc6 [b] | EAc6 [b] |
|---|---|---|---|---|
| | (7 Possible Points) | (25 Possible Points) | (%) | (kWh·Day·20 m$^2$) |
| 90 N. Mary Ave, Sunnyvale | 0 | 19 | 16 | 5.4 |
| 1200 Sierra Point Pkwy, Brisbane | 0 | 22 | 20 | 5.1 |
| 1050 Enterprise Way, Sunnyvale | 0 | 21 | 18 | 5.2 |
| 1620 E. Roseville Parkway, Roseville | 0 | 20 | 17 | 5.3 |
| 10888 White Rock Road, Rancho Cordova | 0 | 24 | 24 | 4.9 |
| 7650 Mission Valley Rd, San Diego | 0 | 14 | 11 | 5.7 |
| 625 N. Mary Ave, Sunnyvale | 0 | 20 | 17 | 5.3 |
| 925 W. Maude Ave, Sunnyvale | 0 | 20 | 17 | 5.3 |
| 220 Jefferson Dr, Menlo Park | 0 | 18 | 15 | 5.4 |
| 6530 Paseo Padre Pkwy, Fremont | 0 | 24 | 24 | 4.9 |
| 800 N. Mary Ave, Sunnyvale | 0 | 21 | 18 | 5.2 |
| 700 E. Middlefield Rd, Mountain View | 2 | 20 | 17 | 5.3 |

Notes: [a] [15]; [b] evaluations performed in the present study.

### 3.2. Evaluating Midpoint Impact and Endpoint Single-Score Damage Results of the $LT_{High}$–$EA_{Low}$ and $LT_{Low}$–$EA_{High}$ Certification Strategies

#### 3.2.1. Midpoint Impact Results

Figure 2 shows the ReCiPe2016 midpoint impact results of $LT_{High}$–$EA_{Low}$ (typical bus), denoted as $LT_{High}$: typical bus, $EA_{Low}$: gas, and of the $LT_{Low}$–$EA_{High}$ (typical and eco-friendly cars) certification strategies, denoted as $LT_{Low}$: typical car, $EA_{High}$: gas and $LT_{Low}$: eco-friendly car, $EA_{High}$: gas, respectively.

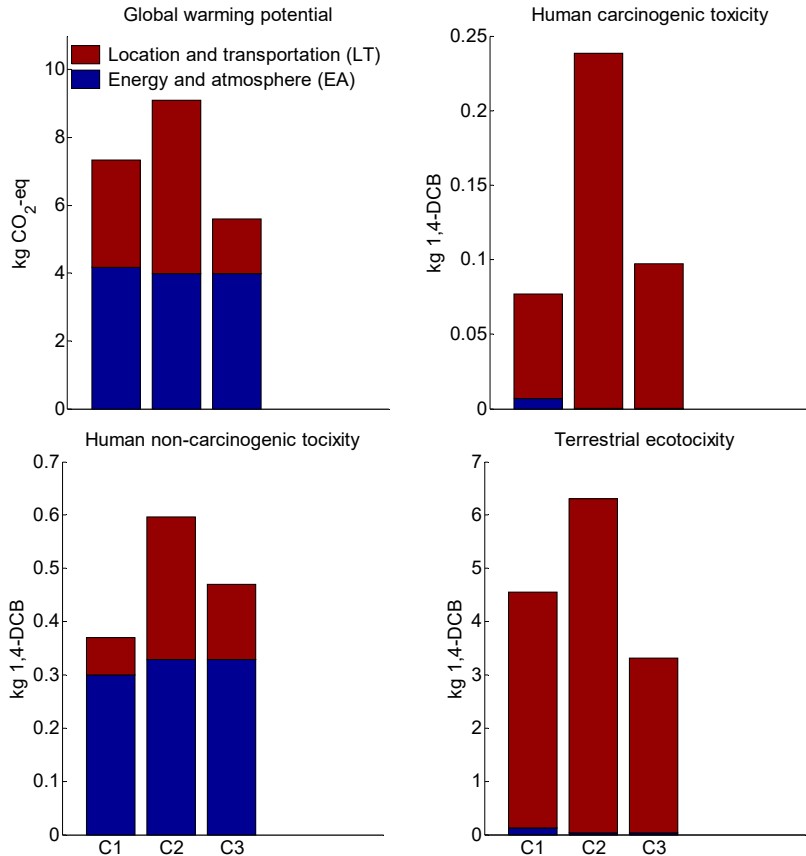

**Figure 2.** ReCiPe2016 midpoint impact results of $LT_{High}$: typical bus, $EA_{Low}$: gas (C1); $LT_{Low}$: typical car, $EA_{High}$: gas (C2); and $LT_{Low}$: eco-friendly car, $EA_{High}$: gas (C3) certification strategies.

It can be noted that $EA_{High}$ and $EA_{Low}$ (OE production) had a high contribution to global warming potential and human noncarcinogenic toxicity, whereas high $LT_{High}$ and $LT_{Low}$ (bus and car transportation) had a high contribution to human carcinogenic toxicity and terrestrial ecotoxicity. Thus, the impact results at the midpoint do not allow us to conclude which of the two processes (OE or transportation) was more influential.

Comparing the impacts of the $LT_{High}$–$EA_{Low}$ and $LT_{Low}$–$EA_{High}$ certification strategies, the following was noted. Analyzing global warming potential, human carcinogenic toxicity, human noncarcinogenic toxicity, and terrestrial ecotoxicity, the $LT_{Low}$: typical car, $EA_{High}$: gas certification strategy was the most environmentally harmful. However, when analyzing global warming potential and terrestrial ecotoxicity, the impact of the $LT_{Low}$: eco-friendly car, $EA_{High}$: gas certification strategy was significantly lower than the impact of the $LT_{High}$: typical bus, $EA_{Low}$: gas certification strategy, whereas, when analyzing human carcinogenic toxicity and human noncarcinogenic toxicity, the impact of the $LT_{High}$: typical bus, $EA_{Low}$: gas certification strategy was much lower than the impact of the $LT_{Low}$: eco-friendly car, $EA_{High}$: gas certification strategy. Thus, based on the midpoint results, it is difficult to determine one preferable certification strategy.

### 3.2.2. Endpoint Single-Score Damage Results

Figure 3 shows the ReCiPe2016 endpoint single-score results of the $LT_{High}$: typical bus, $EA_{Low}$: gas; $LT_{Low}$: typical car, $EA_{High}$: gas; and $LT_{Low}$: eco-friendly car, $EA_{High}$: gas strategies.

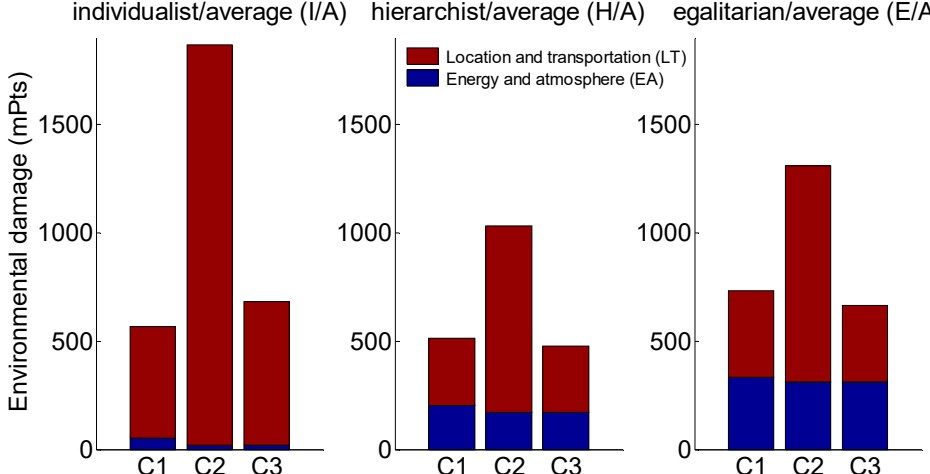

**Figure 3.** ReCiPe2016 endpoint single-score damage results of the $LT_{High}$: typical bus, $EA_{Low}$: gas (C1); $LT_{Low}$: typical car, $EA_{High}$: gas (C2); and $LT_{Low}$: eco-friendly car, $EA_{High}$: gas (C3) certification strategies.

As can be seen, for these certification strategies, transport caused greater damage to the environment than the OE production process. The shares of transport and OE changed, decreasing transport's influence and increasing OE's influence under short, long, and infinite time horizons of pollutants. In particular, in a 20-year period (the I/A option), the transport and OE shares were 91–99% and 1–9%, respectively; in a 100-year period (the H/A option), they were 61–84% and 16–39%, respectively; and in an infinite (1000-year) period (the E/A option), they were 53–76% and 24–47%, respectively.

Comparing the damage involved in the $LT_{High}$–$EA_{Low}$ and $LT_{Low}$–$EA_{High}$ certification strategies, in terms of all three time horizons of pollutants, the $LT_{Low}$: typical car, $EA_{High}$: gas certification strategy that used a typical car was the most environmentally harmful. However, the $LT_{High}$: typical bus, $EA_{Low}$: gas certification strategy that used a typical bus was better than the $LT_{Low}$: eco-friendly car, $EA_{High}$: gas certification strategy, which used an eco-friendly car in a short time horizon (the I/A option), whereas the $LT_{Low}$: eco-friendly car, $EA_{High}$: gas certification strategy was better than the $LT_{High}$: typical bus,

$EA_{Low}$: gas certification strategy in the long (the H/A option) and infinite (the E/A option) time horizons.

Table 9 shows that there was a significant difference between the $LT_{High}$: typical bus, $EA_{Low}$: gas and $LT_{Low}$: typical car, $EA_{High}$: gas certification strategies, as well as between the $LT_{High}$: typical bus, $EA_{Low}$: gas and $LT_{Low}$: eco-friendly car, $EA_{High}$: gas certification strategies. Thus, considering the results presented in Figure 3 and their statistical comparisons presented in Table 9, it can be concluded that the preferability of one strategy over another depended on the time period of the pollutants considered. The $LT_{High}$: typical bus, $EA_{Low}$: gas certification strategy was revealed to be environmentally preferable in the short term, whereas the $LT_{Low}$: eco-friendly car, $EA_{High}$: gas certification strategy was found to be the most environmentally appropriate solution from the long-term or infinite perspectives.

**Table 9.** Statistical evaluation of the ReCiPe2016 endpoint single-score damage results of the $LT_{High}$: typical bus, $EA_{Low}$: gas; $LT_{Low}$: typical car, $EA_{High}$: gas; and $LT_{Low}$: eco-friendly car, $EA_{High}$: gas certification strategies.

| Methodology | Median, 25–75th Percentiles | | | *p*-Value (Cliff's $\delta$) | |
| | $LT_{High}$: Typical Bus, $EA_{Low}$: Gas (C1) | $LT_{Low}$: Typical Car, $EA_{High}$: Gas (C2) | $LT_{Low}$: Eco-Friendly Car, $EA_{High}$: Gas (C3) | C1 versus C2 | C1 versus C3 |
|---|---|---|---|---|---|
| I/A | 567 563—567 | 1871 1870—1871 | 683 682—683 | **0.0000002** (−1.00) | **0.0000002** (−1.00) |
| H/A | 512 495—512 | 1034 1029—1040 | 479 476—485 | **0.0000002** (−1.00) | **0.0001** (0.85) |
| E/A | 734 706—734 | 1311 1302—1314 | 665 656—668 | **0.0000002** (−1.00) | **0.00001** (0.89) |

Notes: *p*-values were evaluated according to three-valued logic; bold font indicates that the value seems to be positive.

## 4. Limitations

In the present study, Spearman's rho ($\rho$) rank-correlation coefficient (effect size) could not be used between the $LT_{LOW}$ and $LT_{High}$ groups because these groups had different sample sizes ($n$ = 12 and $n$ = 14, respectively). Spearman's correlation coefficient can be used to estimate the strength of the monotonic relationship between two LEED credits/categories within one group [38], i.e., within $LT_{LOW}$ or $LT_{High}$. The nonparametric Cliff's $\delta$ was applied to measure the magnitude of the difference between the two distributions (i.e., effect size). Cliff's $\delta$ can be used when there are two independent groups with equal or no equal sample sizes in groups.

## 5. Future Research

Recently, Altomonte et al. [33] used a seven-point Likert scale to assess occupant satisfaction with the indoor environmental quality in LEED-certified buildings (post-occupation analysis). A two-tailed nonparametric Wilcoxon rank-sum test, Spearman's rho ($\rho$) rank-correlation, and Cliff's $\delta$ coefficients were used to calculate significant differences (*p*-value) and substantive significances (effect size) between two independent groups. In the current study, LEED-certified buildings were evaluated using the LEED scorecard (pre-occupation analysis). In the future study, the author plans to compare post-occupation results with pre-occupation results using the above statistical tests.

## 6. Conclusions

This study evaluated the LCAs of two different LEED-CI v4 gold certification strategies for office projects located in cities in California. These two different strategies were revealed by sorting the projects according to the LT category of LEED-CI v4: high and low LT achievements. It was revealed that projects with a high number of LT points performed poorly in the EA category ($LT_{High}$–$EA_{Low}$), whereas projects with a low number of LT points performed well in the EA category ($LT_{Low}$–$EA_{High}$). These two different LEED certification strategies resulted in the same median LEED total score; for the $LT_{High}$ strategy, it was 62.5, and for the $LT_{Low}$ strategy, it was 63.0, resulting in gold certification.

However, from the LCA point of view, the two strategies for obtaining the same LEED certification were quite different. According to the ReCiPe2016 midpoint impact evaluation, the $LT_{Low}$: typical car, $EA_{High}$: gas strategy was the most environmentally harmful certification strategy, whereas, in terms of global warming potential and terrestrial ecotoxicity, the $LT_{Low}$: eco-friendly car, $EA_{High}$: gas strategy was preferable; in terms of human carcinogenic toxicity and human noncarcinogenic toxicity, the $LT_{High}$: typical bus, $EA_{Low}$: gas strategy was the better choice. Thus, on this level of the evaluation, it was impossible to decide on the most environmentally beneficial certification strategy.

According to the ReCiPe2016 endpoint single-score results, the $LT_{Low}$: typical car, $EA_{High}$: gas strategy continued to be the most environmentally damaging certification strategy for all the time horizons of pollutants. However, it was clear that the $LT_{High}$: typical bus, $EA_{Low}$: gas strategy was preferable in the short-term, whereas the $LT_{Low}$: eco-friendly car, $EA_{High}$: gas strategy was preferable from the long-term and infinite perspectives.

The novelty of this study lies in the environmental assessment of the choice of LEED certification strategy. The author has shown that choosing a certification strategy ($LT_{High}$–$EA_{Low}$ or $LT_{Low}$–$EA_{High}$) that results in the same level of LEED-CI v4 (gold) certification resulted in significantly different environmental impacts and damage. Based on the results of the LCAs, it is recommended that LEED certification be carried out with caution in relation to the relevant LCA environmental assessments, thereby increasing the sustainability of buildings.

**Funding:** This research received no external funding.

**Data Availability Statement:** Publicly available datasets were analyzed in this study. The data can be found here: https://www.usgbc.org/projects (USGBC Projects Site) (accessed on 10 April 2022) and http://www.gbig.org (GBIG Green Building Data) (accessed on 10 April 2022).

**Acknowledgments:** The author is grateful to Architect David Knafo for a fruitful discussion of the idea presented in this study.

**Conflicts of Interest:** The author declares no conflict of interest.

## Appendix A

This study used a four-step evaluation procedure in which, for each of the analyzed projects, LEED points were converted into kWh and used as input data for the LCA of $EA_{High}$ and $EA_{Low}$. The procedure included: (1) the conversion of operational energy improvement points to a percentage improvement according to EAc6 [37]; (2) the conversion of 80 kWh·y/m$^2$ to the FU base case, which was 6.4 kWh·day·20 m$^2$; (3) the conversion of the percentage improvement of the FU base case to operational energy saved; and (4) the calculation of the difference between the FU base case and the operational energy saved. Equations (A1)–(A4) give numerical examples of this evaluation procedure for one of the projects located at 235 Pine Street, San Francisco.

$$8 \text{ points of EAc6} = 6\% \text{ operational energy improvement} \tag{A1}$$

$$\text{FU of base case} = \frac{80\frac{\text{kWh·y}}{\text{m}^2}}{250 \text{ days}} \cdot 20\text{m}^2 = 6.4\text{kWh·day·}20\text{m}^2 \tag{A2}$$

$$\text{EAc6 saved operational energy} = 6.4\text{kWh·day·}20\text{m}^2 \cdot 0.06 = 0.4\text{kWh·day·}20\text{m}^2 \tag{A3}$$

$$EA_{Low} = 6.4\text{kWh·day·}20\text{m}^2 - 0.4\text{kWh·day·}20\text{m}^2 = 6\text{kWh·day·}20\text{m}^2 \tag{A4}$$

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
