# Peer review of "Life-Cycle Assessment in the LEED-CI v4 Categories of Location and Transportation (LT) and Energy and Atmosphere (EA) in California: A Case Study of Two Strategies for LEED Projects"

_sustainability, doi:10.3390/su141710893_

Round 1
Reviewer 1 Report
I have read and reviewed the article thoroughly, it is an interesting study, and I think it will be of interest to those working on the same subject. It must be demoted before being published. There is 6% similarity from a single source, plus an overall similarity rate of 21%. After fixing these changes, it can be published.
Author Response
I have read and reviewed the article thoroughly, it is an interesting study, and I think it will be of interest to those working on the same subject. It must be demoted before being published. There is 6% similarity from a single source, plus an overall similarity rate of 21%. After fixing these changes, it can be published.
Author’s answer: The changes were fixed. Line 31-53, 369-380, 492-505.
In addition, in this study, the following subsections “2.4.2. Effect Size Interpretation” and “2.4.3 p-value Interpretation" partially contain texts similar to my other articles. However, this only applies to the description of statistical methods. Changes in the description of statistical methods may lead to erroneous interpretation of the results of the study. I tried to maintain a compromise between the classical description of statistical methods and the reduction of similarity of texts.
For English, please see the attached English Editing Certificate

Reviewer 2 Report
This study presents data analytical investigation for life cycle assessment for LEED certified buildings in California, US. The general approach is interesting, and I appreciated the level of care taken when applying the inferential analyses. I have recommended comparing the effect size threshold to similar EQ studies that have used more stringent thresholds for comparable types of analysis (#10,) but generally, I don’t expect this to significantly change the outcome. There are a few areas of improvement I would recommend, which I have provided, in more detail, below. Firstly, the categorization for LT low and high wasn’t overly clear to me (#7) and I thought this could be easily improved. More importantly, some of the results were not overly easy to understand (#2 and #12.) The larger result appeared to verify the expected outcome, but the more peripheral findings, for other aspects of LEED credits, were not so apparent. These aspects were also not covered in much detail, so they could either be explained more, or removed to make more space for findings more central the aim.
#1: It is worth mentioning that LEED, as applied in v4, does award up to 5-credits to “Building life-cycle impact reduction” (please refer to: Singh, 2017. Whole building life cycle assessment through LEED v4; and LEED v4 BC+D Building life-cycle impact reduction.) These broadly mention some of the general themes raised by the author in section 1.1, with the intent of reuse and optimization of building construction materials.
#2: I would recommending avoid a broad statement that suggests LEED performance is delinked from LCA outcome, considering that for some design credits (e.g. environmental quality), it is not overly clear what this entails, nor how it could be achieved. In fact, I felt that the author raises similar points in P2, L51. Some examples to specific aspects of LEED performance (e.g., credits assigned to materials and resources) could be given instead, which helps explain how LCA is not considered across the long-term performance of the building.
#3: The statistics shown in S1.2 were not overly clear to me. The “score-capacity” by category was not explained, making it unclear what the following values represented; while the monotonically decreased for CV against certification level was difficult to gauge. L72 suggested this relationship goes from platinum through silver and gold, but it might be more accurate to specify this as: silver, gold, and platinum. Further, the problem caused by non-continuous data was also unclear (L79.) The author pinpoints to appropriate solution in the next sentence, overriding any potential issues analysts may encountered from this level of measurement.
#4: P4, L169: The pronoun used consistently was “we”, but unless I am mistaken, there is only one author for this manuscript, and therefore this could be replaced with “I.”
#5: P4, L187: Please specify why it is suitable (e.g., large and diverse enough sample for this state?)
#6: Figure 1. Although the text explaining this figure was well stated, it is not easy to navigate through the plot. The dash line runs on top of important text, and the unnumbered boxes generally don’t provide much insight to the workflow for the method. My suggestions to help improve this plot are to reduce the boxes and text to contain only the most essential details. Any additional unnumbered boxes could be color coordinated with the numbered ones, and placed more carefully alongside the study timeline. This might work better by having a horizontal orientated plot, rather than a vertical one.
#7: Table 1: Please consider reordering the columns from low, intermediate, to high. Unless I also missed this, please also specify how credit achievement was binned into these three categories. Underlying rational would be helpful here. The table caption should also read that this includes both the number of points (credits?) and office projects per category. Further, please specify why some cells have two values, and for some intermediate values (e.g., 14-15) credit achievement equalled those in the high category (e.g., 15-17). This may need to be reorganized to include more accurate ranges (e.g., 4 ≤ intermediate < 17, and high ≥ 17), since in Table 2, LT high and low also have different ranges for the number of points in Table 1.
#8: Table 2: This might be a simple typo; however, the information in this table is not clear. Both headers read the number of points for LThigh and LTlow, but the final row reads the total number of projects, which accumulates to the same number as points. Please indicate also the second column represents the percentage of office occupants that use public transport, or some other indicator.
#9: S2.3.2: The use of the data from Switzerland could be better documented. Some details regarding how comparable and for which transportation parameters this refers to would be useful.
#10: I appreciated the use of the effect size to accompany the inferential tests. Besides these did the author consider verifying the assumption of equal variance? I also noticed that benchmarks for Cliff’s, d, were compared against values lower than other studies. This same statistical procedure, using the same tests, was also applied to LEED EQ data in: Altomonte et al., 2017. Indoor environmental quality and occupant satisfaction in green-certified buildings. It might be worth comparing these to the stricter thresholds used in the above, considering that would be much easier to identify larger effects using the current values. Also, please specify how much data was included, since the authors mentioned that the S-W tests were significant for some cases, but the test itself should only be used for samples less than <50 (see: Mishra et al., 2019. Descriptive Statistics and Normality Tests for Statistical Data.)
#11: P9, L383: Please consider removing this sentence, since the reference doesn’t applied to building science studies (i.e., school psychology research.) There are also, in fact, quite a large number of studies (e.g., please see above) that have utilizing effect sizes and corresponding benchmarks across EQ studies. There is generally no consensus in the field, but my understanding is that conservative thresholds are preferred, considering that it provides more stringent interpretations for data analyses.
#12: Table 5: The results are quite interesting, yet they are not easy to interpret for some categories. While connection between LT values, and LT high and low is clear, this is less so, for example, for EQ and IN, despite these also being significant. The author somewhat mentioned the point that interdependency for credits is related to site selection, so although there are differences that appear in the inferential analyses, it is generally not clear what this could conclude (L427.) For Table 6, results seemed to be more difficult to understanding for EQ. It is not clear how LT is related to some of the credits (e.g., daylighting or thermal comfort.) In the text, the author does not mention these results, so an argument could be made to whether or not they are really needed to be presented.
Author Response
This study presents data analytical investigation for life cycle assessment for LEED certified buildings in California, US. The general approach is interesting, and I appreciated the level of care taken when applying the inferential analyses. I have recommended comparing the effect size threshold to similar EQ studies that have used more stringent thresholds for comparable types of analysis (#10,) but generally, I don’t expect this to significantly change the outcome. There are a few areas of improvement I would recommend, which I have provided, in more detail, below.
Firstly, the categorization for LT low and high wasn’t overly clear to me (#7) and I thought this could be easily improved. More importantly, some of the results were not overly easy to understand (#2 and #12.) The larger result appeared to verify the expected outcome, but the more peripheral findings, for other aspects of LEED credits, were not so apparent. These aspects were also not covered in much detail, so they could either be explained more, or removed to make more space for findings more central the aim.
Author’s answer: Please see answers to #7, #2, and #12 presented below.
#1: It is worth mentioning that LEED, as applied in v4, does award up to 5-credits to “Building life-cycle impact reduction” (please refer to: Singh, 2017. Whole building life cycle assessment through LEED v4; and LEED v4 BC+D Building life-cycle impact reduction.) These broadly mention some of the general themes raised by the author in section 1.1, with the intent of reuse and optimization of building construction materials.
Author’s answer: Line 49-53. The broad statement that suggests LEED performance is delinked from LCA outcome was avoided.
#2: I would recommending avoid a broad statement that suggests LEED performance is delinked from LCA outcome, considering that for some design credits (e.g. environmental quality), it is not overly clear what this entails, nor how it could be achieved. In fact, I felt that the author raises similar points in P2, L51. Some examples to specific aspects of LEED performance (e.g., credits assigned to materials and resources) could be given instead, which helps explain how LCA is not considered across the long-term performance of the building.
Author’s answer: Line 160-174. LCA requirements of Building life-cycle impact reduction (MR credit) was given. Some unquantifiable EQ credits were mentioned. The problem statement was reformulated.
#3: The statistics shown in S1.2 were not overly clear to me. The “score-capacity” by category was not explained, making it unclear what the following values represented; while the monotonically decreased for CV against certification level was difficult to gauge. L72 suggested this relationship goes from platinum through silver and gold, but it might be more accurate to specify this as: silver, gold, and platinum. Further, the problem caused by non-continuous data was also unclear (L79.) The author pinpoints to appropriate solution in the next sentence, overriding any potential issues analysts may encountered from this level of measurement.
Author’s answer: Line 68-88. I focused on the main thing, and the secondary reasoning was removed.
#4: P4, L169: The pronoun used consistently was “we”, but unless I am mistaken, there is only one author for this manuscript, and therefore this could be replaced with “I.”
Author’s answer: The pronoun "we" is replaced by "I" throughout the text.
#5: P4, L187: Please specify why it is suitable (e.g., large and diverse enough sample for this state?)
Author’s answer: The explanation is added in Line 204-206.
#6: Figure 1. Although the text explaining this figure was well stated, it is not easy to navigate through the plot. The dash line runs on top of important text, and the unnumbered boxes generally don’t provide much insight to the workflow for the method. My suggestions to help improve this plot are to reduce the boxes and text to contain only the most essential details. Any additional unnumbered boxes could be color coordinated with the numbered ones, and placed more carefully alongside the study timeline. This might work better by having a horizontal orientated plot, rather than a vertical one.
Author’s answer: Figure 1 (line 229-251) was improved (the dash line runs on top of the text and the unnumbered boxes were deleted). However, the horizontally oriented plot was considered unsuitable.
#7: Table 1: Please consider reordering the columns from low, intermediate, to high. Unless I also missed this, please also specify how credit achievement was binned into these three categories. Underlying rational would be helpful here. The table caption should also read that this includes both the number of points (credits?) and office projects per category. Further, please specify why some cells have two values, and for some intermediate values (e.g., 14-15) credit achievement equalled those in the high category (e.g., 15-17). This may need to be reorganized to include more accurate ranges (e.g., 4 ≤ intermediate < 17, and high ≥ 17), since in Table 2, LT high and low also have different ranges for the number of points in Table 1.
Author’s answer: Line 267-273. Table 1 was reorganized.
#8: Table 2: This might be a simple typo; however, the information in this table is not clear. Both headers read the number of points for LThigh and LTlow, but the final row reads the total number of projects, which accumulates to the same number as points. Please indicate also the second column represents the percentage of office occupants that use public transport, or some other indicator.
Author’s answer: Line 292-295. Table 2 was corrected.
#9: S2.3.2: The use of the data from Switzerland could be better documented. Some details regarding how comparable and for which transportation parameters this refers to would be useful.
Author’s answer: Line 340-366. The data from Switzerland was documented in more details, reflecting its transportation parameters and comparability.
#10:
Q: Besides these did the author consider verifying the assumption of equal variance?
Author’s answer: If it is necessary to measure the statistical difference between two independent groups at small sample sizes and when the normality and equal variance assumptions are not met, then an alternative to the t-test is to use two-tailed exact Wilcoxon-Mann-Whitney non-parametric test [e.g., Bergmann et al 2000].
Q: I also noticed that benchmarks for Cliff’s, d, were compared against values lower than other studies.
Author’s answer: Indeed, Cliff's d has two types of interpretations. According to Ramano et al. (2006) 0.147 (small), 0.33 (medium), and 0.474 (large). According to Vargha and Delaney (2000) 0.11 (small), 0.28 (medium), and 0.43 (large). Historically, LEED data has been assessed using the Romano et al (2006) scale. However, in the present study, the effect sizes of the LEED and LCA-LEED data are presented in tabular form, and these data can be interpreted using the Vargha and Delaney [2000] scale.
Reference:
Romano, J.; Corragio, J.; Skowronek, J. Appropriate statistics for ordinal level data: Should we really be using t-test and Cohen’s d for evaluating group differences on the NSSE and other surveys? In Proceedings of the Annual Meeting of the Florida Association of Institutional Research, Cocoa Beach, FL, USA, 1–3 February 2006; Florida Association for Institutional Research: Cocoa Beach,
FL, USA, 2006; pp. 1–33.
Vargha, A. and H.D. Delaney. A Critique and Improvement of the CL Common Language Effect Size Statistics of McGraw and Wong. 2000. Journal of Educational and Behavioral Statistics 25(2):101–132.
Q: This same statistical procedure, using the same tests, was also applied to LEED EQ data in: Altomonte et al., 2017. Indoor environmental quality and occupant satisfaction in green-certified buildings. It might be worth comparing these to the stricter thresholds used in the above, considering that would be much easier to identify larger effects using the current values.
Author’s answer: Line 653-673. Please see the answer in 5. Limitations and 6. Future Research sections.
Q: Also, please specify how much data was included, since the authors mentioned that the S-W tests were significant for some cases, but the test itself should only be used for samples less than <50 (see: Mishra et al., 2019. Descriptive Statistics and Normality Tests for Statistical Data.)
Author’s answer: For LTHigh: typical bus data, in each perspective (i.e., I/A, H/A, and E/A), the Shapiro–Wilk test results showed that the assumption of normality was not met ( = 0.0008, sample size (n), n = 14, in all three methodological options), while for LTLow: typical car and LTLow: eco-friendly car data in each perspective (i.e., I/A, H/A, and E/A), the Shapiro–Wilk test results showed that the assumption of normality was met ( = 0.0598, = 0.1264, and = 0.2222, n = 12, in all three methodological options), respectively. In this context, if one of the two groups does not have a normality assumption, the nonparametric exact WMW test and Cliff’s effect size are used to estimate the statistical difference between the two groups.
#11: P9, L383: Please consider removing this sentence, since the reference doesn’t applied to building science studies (i.e., school psychology research.) There are also, in fact, quite a large number of studies (e.g., please see above) that have utilizing effect sizes and corresponding benchmarks across EQ studies. There is generally no consensus in the field, but my understanding is that conservative thresholds are preferred, considering that it provides more stringent interpretations for data analyses.
Author’s answer: The sentence was removed. Please see the answer in Line 426-430.
#12: Table 5: The results are quite interesting, yet they are not easy to interpret for some categories. While connection between LT values, and LT high and low is clear, this is less so, for example, for EQ and IN, despite these also being significant. The author somewhat mentioned the point that interdependency for credits is related to site selection, so although there are differences that appear in the inferential analyses, it is generally not clear what this could conclude (L427.) For Table 6, results seemed to be more difficult to understanding for EQ. It is not clear how LT is related to some of the credits (e.g., daylighting or thermal comfort.) In the text, the author does not mention these results, so an argument could be made to whether or not they are really needed to be presented.
Author’s answer: Line 465-471. The influence of the site selection category on the WE, EA, MR and EQ categories was clarified (identified by Ismaeel [36]).
I decided not to present the EQ and IN credits (please see Line 474-479). As a consequence, EQ credits were removed from Table 6.
For English, please see English Editing Certificate.

Reviewer 3 Report
The topic of the paper entitled Life-Cycle Assessment in the LEED-CI v4 Categories Location and Transportation (LT) and Energy and Atmosphere (EA) in California: A Case study of Two Strategies for LEED Projects” is interesting and worthy of investigation. Also, this paper is well written, including the introduction, Materials and Methods, Results, and conclusions. However, the reviewer has some recommendations that thinks will improve a future version of this work.
- Abstract: At the end of the abstract, one or two sentences should be added as conclusions and recommendations to give a complete picture of the content of this paper.
- Material and Methods: 1) Figure 1 should improve more smoothly for the reader with the use of the appropriate/right font and color. 2) in Table 2, sources a and b should add as references [1,2], unacceptable copy and paste only the website of the information. 3) Table 3, references should add to the section of data source in the table. 4) Table 4, I did not understand kg CO2eq!! do authors mean kg CO2eq/GW or what, please explain, because authors talk here about operational energy (OE) processes. Moreover, all tables need to be properly reformatted and revised.
- Results: Please clearly improve this section according to the above comments, i.e., explain and justify the key findings with clear figures. In other words, please improve the quality of results section by focusing on key messages. Also, refine Figures 2 and 3 better.
- Conclusions: Limitations of this study and potential future development of this research should also be included in the manuscript.
Author Response
The topic of the paper entitled Life-Cycle Assessment in the LEED-CI v4 Categories Location and Transportation (LT) and Energy and Atmosphere (EA) in California: A Case study of Two Strategies for LEED Projects” is interesting and worthy of investigation. Also, this paper is well written, including the introduction, Materials and Methods, Results, and conclusions. However, the reviewer has some recommendations that thinks will improve a future version of this work.
- Abstract: At the end of the abstract, one or two sentences should be added as conclusions and recommendations to give a complete picture of the content of this paper.
Author’s answer: Line 23-25. The concluded sentence was added.
- Material and Methods: 1) Figure 1 should improve more smoothly for the reader with the use of the appropriate/right font and color. 2) in Table 2, sources a and b should add as references [1,2], unacceptable copy and paste only the website of the information. 3) Table 3, references should add to the section of data source in the table. 4) Table 4, I did not understand kg CO2eq!! do authors mean kg CO2eq/GW or what, please explain, because authors talk here about operational energy (OE) processes. Moreover, all tables need to be properly reformatted and revised.
Author’s answer: 1) line 229-251. Figure 1 was improved. 2) Line 296. In Table 2, sources a and b were added as references [14] and [15], respectively. 3) Line 334. The reference was added. 4) Line 385 and 386. CO2 eq was explained. This impact is relevant to both transportation and operational energy processes. In addition, tables 1,2, and 5 were revised and reformatted.
- Results: Please clearly improve this section according to the above comments, i.e., explain and justify the key findings with clear figures. In other words, please improve the quality of results section by focusing on key messages. Also, refine Figures 2 and 3 better.
Author’s answer: Line 465-471. The influence of the site selection category on the WE, EA, MR and EQ categories was clarified (identified by Ismaeel [36]).
I decided not to present the EQ and IN credits (please see Line 474-479). As a consequence, EQ credits were removed from Table 6.
Figure 2 and 3. LT was replaced with Location and Transportation and EA was replaced with Energy and Atmosphere.
- Conclusions: Limitations of this study and potential future development of this research should also be included in the manuscript.
Author’s answer: Line 653-673. 5. Limitations and 6. Future Research sections were included.
Round 2
Reviewer 2 Report
Thank you for taking all my comments into consideration. These changes were on-point and have been clearly reflected by the tracked-changes found in the amended manuscript. Some minor comments, which are mainly presentation based and not content related, have been provided for further consideration:
#1: Please check the formatting for Table 1. I think this may have been an unfortunate editing problem that misaligned the table rows and columns. Please consider amending this for the final version.
#2: The cumulative values in Table 2 were still not apparent to me (please see original comment #8.) If columns three and four are the points achieved, it is unclear how this is equal to the total number of LEED projects in the final row. Unless I am mistaken, I think the final row should read as the total number of points achieved.
#3: Please consider placing the limitations and future study sections before the conclusions, since these should not overshadow the main points the author has found from their research endeavor.
Author Response
Q#1: Please check the formatting for Table 1. I think this may have been an unfortunate editing problem that misaligned the table rows and columns. Please consider amending this for the final version.
Ans#1: I have corrected the format of the rows and columns of the Table 1 (line 272-273).
Q#2: The cumulative values in Table 2 were still not apparent to me (please see original comment #8.) If columns three and four are the points achieved, it is unclear how this is equal to the total number of LEED projects in the final row. Unless I am mistaken, I think the final row should read as the total number of points achieved.
Ans#2: Table 2 is a continuation of Table 1. Table 2 is devoted only to the Gold group. The gold group consists of two subgroups: LTLow (1-3 points achieved) and LTHigh (17-18 points achieved). The third and fourth columns of the Table 2 show the number of LEED projects in different California cities, but not the points achieved.
#3: Please consider placing the limitations and future study sections before the conclusions, since these should not overshadow the main points the author has found from their research endeavor.
Ans#3: I have moved the limitations and future study sections before the conclusions (line 627-645)
This manuscript has been double checked by professional proofreaders. These checks were carried out through the English language editing by MDPI system.
Reviewer 3 Report
Compared to the first draft of this manuscript, this revised manuscript has been improved by addressing the reviewers' comments.
Author Response
The reviewer's comment:
Compared to the first draft of this manuscript, this revised manuscript has been improved by addressing the reviewers' comments.
The author's replay:
Thank you very much.